# Eukaryotic Microalgae Communities from Tropical Karstic Freshwater Lagoons in an Anthropic Disturbance Gradient Microscopic and Metagenomic Analysis

**DOI:** 10.3390/microorganisms12112368

**Published:** 2024-11-20

**Authors:** Vanessa Rosaldo-Benitez, Gerardo A. Ayil-Chan, Natalia Labrín-Sotomayor, Ruby Valdéz-Ojeda, Yuri J. Peña-Ramírez

**Affiliations:** 1El Colegio de la Frontera Sur Unidad Campeche, Sustainability Sciences Department, San Francisco de Campeche 24000, Mexico; vanessa.rosaldo@posgrado.ecosur.mx (V.R.-B.); gerardo.ayil@posgrado.ecosur.mx (G.A.A.-C.); nlabrin@ecosur.mx (N.L.-S.); 2Centro de Investigación Científica de Yucatán AC, Renewable Energies Unit, Mérida 97205, Mexico; ruby.valdez@cicy.mx

**Keywords:** microalgae diversity, water quality, bioindicator, functional genomics, rare microalgae

## Abstract

The Yucatan aquifer sustains the people living in the Mayan forest and its associated fauna. Human activities threaten water quality and the environmental services associated with it. To assess the eukaryotic microalgae community structure as a bioindicator of water quality, we employed a combined approach of microscopic and shotgun metagenomics to identify specific genera associated with shifts in physicochemical parameters in three permanent lagoons located in Campeche, Mexico. We could identify highly complex and diverse communities independent of human activity intensity, harboring an average of 362 genera at each lagoon. Of those, 85 were affected by alkalinity, carbonates, water hardness, and cyanuric acid levels. Some genera, like *Nannochloropsis* and *Thraustotheca*, showed significant negative correlations with lead concentration. The functional annotation of genes revealed these communities’ highly diverse metabolic capabilities and the pending work for extensive genomic characterization of rare clades.

## 1. Introduction

The current scenario of contamination of wetlands in the tropics has generated interest in knowing the specific causes that negatively or positively influence the state of health of aquifers [1]. The use of physicochemical parameters as the sole input to determine the health of water bodies is insufficient to evaluate the effects of human activities since it is not possible to show the impact that various factors can have on the biological communities that depend on it, such as microbial communities, including bacteria, microalgae, and zooplankton [2]. Moreover, ecosystem deterioration affects the rest of the organisms, including those species that feed people [3,4]. In addition to the physicochemical parameters, microbial communities emerge as a relevant indicator of some specific taxa, like Diatoms, and can be sensitive to changes in the environment, like variations in pH, organic matter, or dissolved oxygen, altering the reproduction of these microorganisms or causing teratologies [5,6]. Changes in the microalgae population structure due to disturbances in their habitat give them a widely recognized value as bioindicators of water health. Some genera present in continental water bodies that are suitable bioindicators belong to the microalgae genera *Chlorella*, *Scenedesmus*, *Ankistrodesmus*, *Cosmarium*, and *Coelastrum*, in addition to Diatoms such as *Navicula* [7] and several genera of cyanobacteria like, *Aphanizomenon*, *Chroococcus*, *Cuspidothrix*, *Cyanodyction*, *Dolichospermum*, and *Microystis* [8,9,10].

Usually, microscopic observation is employed to identify and quantify microorganisms. However, this technique has limitations due to its low sensitivity and specificity [11]. On the other hand, recent emerging tools, such as metabarcoding, have been used to elucidate microalgal community structure [11,12]. Unfortunately, microalgae are too diverse. When choosing a genetic marker, the results can be biased because only specific groups can be identified [13]; in contrast, if a universal marker is selected, information can be obtained from highly conserved regions. There is a risk of confusing taxa at the species or even genus level [14]. As an alternative, shotgun sequencing allows the obtainment of information on the entire genome, with which the communities’ taxonomic structure and functionality can be elucidated. However, existing databases for groups such as eukaryotic microalgae are limited for freshwater taxa, being more robust for saltwater species [15]. Although both identification methods (microscopy and metabarcoding or metagenomics) have advantages, the combined use of both approaches is very useful in complementing and providing more significant information on microalgal community structure [16].

To assess the diversity and abundance of eukaryotic microalgae that could be used as perturbation indicators, we combined microscopic and shotgun metagenomic analysis to evaluate three permanent water bodies in a karstic environment in the Yucatan Peninsula, Mexico. In this region, continental karstic wetlands predominate, such as natural sinkholes (locally known as “*cenotes*” from the Mayan “*tzonot*”), rivers, and permanent and temporary lagoons [17]. Unfortunately, anthropogenic activity in this region has deteriorated their hydrological, biological, and ecological functions [18]. Most lagoons, *cenotes*, or rivers in this region present some degrees of anthropogenic impact, ranging from preserved areas located in protected reserves without visible anthropogenic effects on those affected by multiple contiguous human settlements with intensive resource exploitation and degraded surrounding forests by land use change associated with livestock and agriculture [19,20]. These contrasting conditions are particularly interesting as models to evaluate whether disturbance can affect or favor the presence of eukaryotic microalgae with putative bioindicator characteristics. In this work, we evaluated three lagoons of the Yucatan aquifer with different degrees of anthropic impact. We determined the diversity and abundance of eukaryotic microalgae by combining microscopic observation and shotgun metagenomic DNA sequencing to identify the existing communities in each condition. Finally, we analyze the effect of physicochemical conditions at each sampling site with the presence or abundance of each taxonomic group found. According to the impact of anthropization has shown on eukaryotic microalgae, we expected to detect shifts in taxa sensitive to eutrophication like Diatoms [21] and other groups like mixotrophs [22].

## 2. Materials and Methods

### 2.1. Study Sites and Physicochemilcal Parameters Determination

For this work, we evaluated three lagoons in Mexico’s southern part of the Yucatan Peninsula. This region is a karstic ecosystem harboring an underground aquifer. Lagoons in this region are primarily shallow, fed by runoff seasonal rain (1200 to 1800 mm y^−1^) lasting from June to November. A semi-deciduous tropical forest covers the land surface with a dominant Aw climate, where more than 500K ha (30% surface) has been declared natural reserves. The three lagoons considered in this study are permanent, suffering seasonal variations no higher than 20% in depth and extension [23]. Sampling take place in October 2023 in the *Mocu* lagoon, in the *Champotón* Municipality (area voluntarily designated for conservation since 2018 considered pristine) (18.78 N; 90.50 W); in the *X-Canha* lagoon, located in the *Hopelchén* municipality (semi-disturbed, area destinated to recreation situated in the buffer area of the Calakmul reserve) (19.10 N; 89.29 W); and the *Silvituc* lagoon, located at the *Escárcega* municipality (disturbed, area, primarily devoted to agriculture with patches of tropical forest) (18.64 N; 90.27 W) (Figure 1). A total of 12 composite samples were collected for this study. Each composite sample corresponded to a mix of seven independent water samples. Sampling was performed at two representative sampling points in each lagoon: the dock and the lagoon centroid next to the dock. At each sampling point, two samples were taken, one at the surface and the other at the bottom. Approximately 3.5 L of water was collected from each sampling site using seven consecutive 500 mL aliquots obtained using a Van Dorn bottle directly transferred to a sterile plastic container. Between samples, the Van Dorn bottle was sanitized using 0.1 N HCl and rinsed three times with sterile water. Each composite sample was used for microscopic observation and metagenomic analysis. One liter of sample was also collected to measure chlorophyll-a concentration (see next section). The collected samples were labeled, wrapped in aluminum foil, placed inside black bags, and placed in ice for immediate transportation to the laboratory. Transparency at the evaluated depth was measured with a Secchi disk. Additionally, some physicochemical parameters were measured three times in situ such as temperature, pH (Instrument Cat. HI98127, Hanna instruments Woonsocket RI, USA); dissolved oxygen (Instrument Cat. DO9100 Rcyago, Shenzhen Yage Technology Limited, Shenzhen, China); conductivity, salinity, total dissolved solids (TDS) (Instrument Cat. EC500 Extech, Nashua, NH, USA), for continuous values, we employed a parametric *t* test to determine significant differences (*p* < 0.05). Water hardness, total alkalinity, fluoride, cyanuric acid, carbonates, free and total chlorine, bromide, nitrates, nitrites, iron, chrome, lead, copper and mercury were determined using commercial test strips (Varify Co. San Diego, CA, USA). For strip-determined values, three independent determinations were taken and the concentration range was obtained according the color reference, for these values a non-parametric test (Kuskal–Wallis) was applied to determine differences between samples (*p* < 0.05), using a zero value for those samples with non-detectable levels. Finally, the online calculator of photic zone depth and attenuation coefficient was employed to determine those values (https://iim.unah.edu.hn/grupos/giica/calculadora-zeu/ (accessed on 2 June 2024)). The Pearson correlation test was employed to determine statistical significances within physicochemical parameters and relative abundance of found taxa (*p* < 0.05).

### 2.2. Chlorophyll-A Determination

One liter of water per sample was filtered at atmospheric pressure using a vacuum pump in the dark using fiberglass filters (Whatman 47 Ø mm/0.7 µm, Whatman plc. Maidstone, UK), where all possible moisture was extracted. They were then cut to place them in 15 mL Falcon tubes, where 10 mL of 90% (*v/v*) acetone was added to each filter and shaken for 1 min with the help of vortex and refrigerated at 4 °C in the dark for 24 h. Subsequently, the samples were centrifuged at 3000 g for 30 min. Using a spectrophotometer (Thermo Scientific™ Multiskan GO™ with Skanlt^TM^ Software for Microplate Readers. v1.0 2017 Whaltham, MA, USA), the absorbance was measured at 664 nm and 750 nm before acidifying followed by a determination at 666 nm and 750 nm 90 s after acidifying with 100 µL of 0.1 N HCl. The data obtained were applied to the following formula to obtain the concentration of chlorophyll-a [25]:$${Cl}_{\alpha }(µg/L)=\frac{A*K\left(Abs664b-Abs666a\right)V\;Acet}{Vf*t}$$
where

A: Chlorophyll-a absorbance coefficient = 11.0

K: Rate that expresses the correction for acidification = 2.43

664b: Absorbance at 664 nm before acidifying

666a: Absorbance at 666 nm after acidifying

V Acet: Volume of acetone used in the extraction (mL)

Vf: Volume of filtered water in L

t: Optical cell path in cm

### 2.3. Microscopic Observations

The samples were centrifuged at 20,000× *g* for 20 min (100Y rotor cat. 75,004 Thermo Scientific with Fiberlite™ F15-6, Whaltham. MA, USA), and part of the recovered biomass (100 μL) was used for microscopic observation (Carl ZEISS, mod. Primo Star with ZEN 2.5 blue edition Software, Oberkochen, Germany) at 100×, where dilutions of 1:10,000 were made. An amount of 10 µL was placed on a slide for observation with the help of the microscope at 100×, using a catalog of freshwater microalgae to review phenotypic characteristics and identify the microalgae [26,27,28]. Based on identification, a qualitative incidence table was generated for further analysis (Appendix A).

### 2.4. Metagenomic Analysis

A measurement of 100 mg of biomass from each sample obtained from pellets from the previous section was used for DNA extraction, following the working sequence of the Zymo Plant/Seed Miniprep Cat kit #D6020 following the manufacturer’s instructions. Subsequently, a spectrophotometer was used to measure quality and concentration (Thermo Scientific™ Multiskan Go™ with Skanlt Software for Microplate Readers Version 1.0 2017) and agarose agar 0.8% for integrity. After passing all quality tests, the samples were sent to an external Shallow Shotgun Metagenomic Illumina-based Sequencing service at a Novogene Corporation.

### 2.5. Bioinformatic and Statistical Analysis

The sequences received as *.fastq files were processed on the KBase platform [29] (http://www.kbase.us/ (accessed on 2 June 2024)), using paired library objects to assemble reads using metaSPADES v3.15.3 [30] (available online: https://github.com/ablab/spades (accessed on 2 June 2024)). Taxonomic assignation was performed using Kaiju v1.9.0 [31] (http://kaiju.binf.ku.dk/ (accessed on 2 June 2024)), employing the NCBI database for Eukaryotes + Prokaryotes. After eukaryotic microalgae taxa filtering, gene annotation was performed by EggNOG mapper v2.1.9 [32] using the DIAMOND algorithm [33] (https://github.com/eggnogdb/eggnog-mapper/wiki/ (accessed on 2 June 2024)). To obtain ecological indices, ordination analysis, Kruskal–Wallis, and hypothesis tests, we employed the PAST v4.17 and Infostat v2020 software [34].

## 3. Results

### 3.1. Physicochemical Differences Among Lagoons

We determined water physicochemical parameters to evaluate the water quality at each water body. The lagoons are shallow, with a maximum depth of 2.5 m; temperate, averaging water temperatures of 30.9 °C; and alkaline, with pH values between 7.5 and 8.5. Besides these apparent similarities, statistical differences were found between the lagoons for all evaluated parameters (*p* < 0.05) (Table 1 and Table 2). Concerning the depth in the column water where samples were taken, the samples from the surface showed differences for de *X-Canchá* lagoon only for chlorophyll-a content; however, neither photic layer nor attenuation coefficient values show any correlation with chlorophyll-a content (R = 0.15, R = 0.04, respectively). According to our data, we could not detect a pattern in physicochemical values that correlates to the expected differences in the anthropization gradient (Table 3).

### 3.2. Microscopic Identification of Microalgal Communities

Our data revealed the presence of 8 phyla, 16 classes, 27 orders, 47 families, and 62 genera (Appendix A) distributed in distinct communities in each lagoon sampled. Of the 62 genera identified in our samples, 35 were found in the *Mocú* lagoon (Shannon H’ = 3.89), 13 in the *X-Canhá* lagoon (Shannon H’ = 2.97), and 46 in the *Silvituc* lagoon (Shannon H’ = 4.07). Seven genera (*Anabaena*, *Asterocapsa*, *Chlamydomonas*, *Cosmarium*, *Dictyococus*, *Gomphosphaeria*, and *Leptolyngbya*) were identified in all the lagoons (Figure 2a). Differences were also found between sampling areas, where 43 genera were found exclusively on the surface samples. In contrast, only *Frustulia* was exclusive from the depth bottom samples, resulting in 21 genera present in surface and bottom samples. Clustering analysis based on Bray–Curtis similarity did not show any grouping pattern by lagoon or depth (Figure 2b), and even an ordaining analysis by NMDS showed separated polygons (Figure 2c). The PERMANOVA test did not reveal significant differences. However, the SIMPER test revealed that 15 genera (*Ankistrodesmus*, *Tetraedron*, *Oocystis*, *Anabaena*, *Arthrospira*, *Leptolyngbya*, *Surirella*, *Phacus*, *Pandorina*, *Chlamydomonas*, *Ulnaria*, *Merismopedia*, *Klebsormidium*, *Euglena*, and *Cosmarium*) out of 62, are responsible for 50.91% of the sample differences. A photographic compilation of representative genera per lagoon is presented in Figure 3.

### 3.3. Metagenomic Identification of Microalgal Communities

We obtained shallow sequence data (available through the BioProject PRJNA1162095; https://www.ncbi.nlm.nih.gov/bioproject/1162095 (accessed on 4 September 2024) (publicly available upon publication) processing yielded 79.16 million reads, equivalent to 11.87 Gpb, averaging 150 pb per read, and a quality score mean of 36.11 on the Phread scale. Once the taxonomic assignation concluded, those taxa representing eukaryotic microalgae were filtered, yielding 28 phyla, 72 classes, 211 orders, 318 families (or equivalent clades), and 518 genera. Of these 518 genera, only 10 were detected also by microscopy (*Chlamydomona*, *Chlorella*, *Chlorococcum*, *Desmodesmus*, *Dictyococus*, *Euglena*, *Glaucocystis*, *Peridinium*, *Scrippsiella*, and *Vischeria*). Our metagenomic-derived data revealed the presence of 315 genera in the *Mocú* lagoon (Shannon H’ = 5.46; Simpson 0.992), 184 genera in the *X-Canhá* lagoon (Shannon H’ = 4.94; Simpson 0.989), and 438 genera in the *Silvituc* lagoon (Shannon H’ = 5.59; Simpson 0.993). (Appendix A). Rarefaction analysis of microalgae genera revealed that our sampling effort in the *Silvituc* lagoon captured 88% of the expected taxa, 71% in the *Mocú* lagoon, and 42% in the *X-Canhá* lagoon (Appendix A). However, using the metagenomic-based approach, we identified 165 genera common to all three lagoons (Figure 4a). The clustering analysis of samples by Bray–Curtis similarity did not show any grouping by lagoon site or sampling point, separating only the sample obtained in the *Silvituc* lagoon, center-bottom (Figure 4b). The relative abundance of microalgae revealed that 49.87% of the obtained genera belong to unknown eukaryotic taxa, whereas at the phylum level, the unknown OTUs are reduced to 0.79%. Except for one sample from the *Silvituc* lagoon, all samples showed a similar microalgal community dominated by genera *Euglena*, *Symbiodinium*, *Desmodesmus*, *Chlorella*, *Hydrodictyon*, *Volvocaceae*, *Nannochloropsis*, *Cryptomonas*, *Ankistrodesmus*, and *Eustigmatophyceae* (Figure 4c). The *Silvituc* sample obtained from the bottom was the most contrasting with respect to the other samples; in this case, the microalgal community was widely dominated by *Ptilothamnion*, *Thraustotheca*, *Haramonas*, *Micromonas*, and *Cryptomonas*, even though the biodiversity indices remain high (Shannon H’ = 4.27; Simpson = 0.97). For all samples, however, the genera below 1% of microalgae relative abundance averaged 52.04% ± 5.78% (standard deviation) of the total diversity. The PERMANOVA test did not show significant differences between lagoons or sampling sites. However, NMDS analysis of samples yielded a clear ordination of samples by lagoon site (Figure 4d). SIMPER test revealed that 70 out of 518 genera (Appendix A) are responsible for 50% of the differences between the lagoons. Following these criteria, only the common genera *Euglena*, *Tetraedron*, and *Vischeria* detected with both microscopy and metagenomics, were included in the SIMPER list.

### 3.4. Correlation Analysis

To identify those taxa representatives susceptible to environmental variations associated with sampling sites, microbial eukaryotic communities, including microalgae, were analyzed by Pearson correlation (Figure 5). Those genera (15 identified by microscopy and 70 with metagenomics) contributing up to 50% to the differences between lagoons obtained by SIMPER were considered in this analysis. We found that most detected genera inversely correlated with alkalinity, cyanuric acid, carbonate, and lead concentrations. However, significant correlations were found for *Nannochloropsis* and *Trebouxia* that correlated negatively with lead concentrations. Significant positive correlations were more common, being nitrites, nitrates, and chlorine factors that correlated positively with genera like *Auxenochlorella*, *Chamydomona*, *Chlorella*, *Desmodesmus*, *Dictyopteris*, *Euglena*, *Hydrodyction*, *Merisimopedia*, *Synura*, and *Volvocaceae*. The pH and the dissolved oxygen concentrations were negatively correlated with the presence of *Ankistrodesmus* and *Arthrospira* genera, whereas water hardness appears to significatively affect the presence of genera *Merismopedia*, *Surinella*, and *Tetraedron.*

### 3.5. Functional Analysis

To unveil the metabolic capabilities of sampled microalgal communities, EggNOG-derived functional annotation, we obtained 70 functions clustered by ortholog gene function (COGs), resulting in 17 functional categories (Figure 6). The more abundant cluster accumulated 26.8% of functions classified in the S category corresponding to unknown functions, followed by 8.1% assigned to the O category, corresponding to post-translational modification, protein turnover, and chaperone functions, followed by COGs K, transcription, and G, carbohydrate metabolism and transport with 7.3% each. Samples showing higher functional diversity were those collected in the *Mocú* lagoon with 15 COGs, whereas samples from the *X-Canhá* lagoon showed only 5 COGs. The sample from the bottom of the *Silvituc* lagoon was represented exclusively by COG functions involved in translation mechanisms with specific functions related to ribosomal protein S31e, translation–initiation factor 2, and eukaryotic translation–initiation factors.

## 4. Discussion

### 4.1. Physicochemical Properties in Sampled Lagoons

Maintaining good water quality in continental freshwater bodies has become a priority for maintaining healthy ecosystems. Excessive dry or rainy seasons associated with global climate change and human population growth increase the pressure on these bodies, contributing to the freshwater crisis. Karstic ecosystems are particularly vulnerable to those impacts due to the permeability and connectivity of underground aquifers, which are prone to transport contaminants from relatively long distances. Of particular interest in rural areas are those anthropic contaminants linked to agriculture and livestock activities. Fertilizers and heavy metals derived from agrochemicals contribute to eutrophication. The model site we chose for this work, the Campeche State in Mexico, harbors the most extensive natural reserves, both in number and size. In this region, we sampled three permanent lagoons with different degrees of perturbance. Our data, however, showed that physicochemical values did not entirely reflect the expected behavior. We found transparency, salinity, total dissolved solids, bromide, iron, and lead levels similar to an aquifer considered contaminated; these values for the *Mocú* lagoon, which is surrounded by at least 10 Km of highly conserved forest, would be susceptible to receive inputs from the closest biggest city (*Escárcega* city, ≈30,000 habitants, 28 Km away), or the closest semi-intensive monoculture area (*Justicia Social* locality, ≈6500 Ha, 26 Km away).

Moreover, salinity levels in the *Mocú* lagoon may suggest saline water intrusion from the coast (60.55 Km away). This is consistent with the permanent counter-flow movement of seawater intrusion in the north side of the Yucatan Peninsula up to 80 Km away [28,29] and would explain the presence of lead due to the permanent high-scale petroleum-extracting activities in the Campeche Bay (≈70 Km away from the coastline). However, no seawater intrusion was previously detected in the southern Campeche state [35]. On the other hand, the “*Nortes*” events, locally called the windy and punctual rainy storms linked to cold air waves coming from the north and hitting the coastline during January and February, could indeed affect lagoons relatively close to the coast, by altering the halocline in the underground aquifer. “*Nortes*” then may also explain the seawater intrusion in the *Mocú* lagoon [36].

In contrast, the *Silvituc* lagoon, surrounded by human settlements, has a low salinity presence that discards seawater intrusion. Previous work on this lagoon has demonstrated high sulfate levels associated with rain and runoff [35]. In summary, the surrounding conservation state of forest coverture appears to be unrelated to all water quality parameters, even though contrasting conditions were found between sampled lagoons, which may modulate microalgae community structure in these sites. This means that the different degrees of the anthropization state of the surrounding environment in sampled lagoons did not impact water quality. This could suggest that human impact in these zones is low enough to be neutralized or removed by the natural water the interchange process.

### 4.2. Eukaryotic Microalgal Communities

Our study identifies 570 genera (or equivalent clade) of eukaryotic microalgae in the sampled lagoons. This is, by far, the highest number of taxa previously identified in this region. The high diversity index values obtained reveal that the three sites harbor a vast diversity of taxa. Our rarefaction analysis results reflected the existing potential for more profound studies in this region as a source of novel microalgae, as the identified genera ranged only 50% of OTUs belonging to these eukaryotic clades, and our effort covered only an average of 67% of the expected genera taxa present on these sites. In addition to the different successes in taxa capturing, our data reveal the consistent microalgae community structure, except for the sample obtained from the bottom of the *Silvituc* lagoon, even though this site showed a microalgae richness similar to the rest of the samples and that the physicochemical levels were similar in this sampling site. In this case, the community structure shift may be due to an elusive factor such as competing bacteria, predatory microorganisms, or some taxa-specific deleterious factor. Beyond this particular case, the eukaryotic microalgae community did not reflect any pattern in response to the surrounding environment anthropization. This is congruent with our physicochemical analysis results, which support the idea that these lagoons are resilient enough to human impact in these zones.

An increasing number of microalgae taxa are being identified and recorded on international databases, fueled mainly by the rising interest in biotechnology models [37]. Even karstic environments are often dominated by *Chlorophyceae*, *Trebouxiophyceae*, *ex-Diatomophyceae* (*Bacillariophyta phylum*) families [12,38,39]; these and other less common taxa become interesting for bioprospection, particularly as bioindicators for anthropic activities [40]. Interestingly, most of the genera contributing to >50% of differences between lagoons by SIMPER analysis responded negatively to the presence of alkalinity, cyanuric acid, carbonates, and lead. In general, high alkalinity and carbonates are common in karstic aquifers, inhibiting the growth of several microalgae genera [9,35], which suggests that the abundance of these taxa may infer the differences in water quality in sampled lagoons.

The combined strategy for microalgae identification using microscopy and shotgun metagenomics resulted in exciting evidence. As has been previously reported in works where a similar approach was employed [41,42], there exists a considerable gap at deeper taxonomic levels necessary to compare at the phylum (or equivalent clade) level mainly due to the high similarity of phenotypic characteristics complicating optical identification of some taxa [43]. These differences have also been assumed by the presence of extracellular DNA that can only be detected by molecular tools [41]; however, in our case, samples for microscopic and metagenomic analyses were purified by centrifugation instead of filtration to avoid the presence of such contaminants and most bacteria and virus. However, the presence of dead microalgal carcasses, usually discarded for microscopic analysis, could carry DNA suitable to be detected for the metagenomics approach.

Climatic change has been evaluated in the Yucatan Peninsula and Central America, and a consistent increase in temperature in the last 50 years [44] has altered the rainfall regime. Events like the El Niño–Southern oscillation and the Atlantic multidecadal oscillation cause a high interannual variability and long-term increase in the rainy season duration [45]. These climatic alterations, plus the strong deforestation rates and anthropization, may finally disrupt the resilience of microalgae communities, potentially affecting regulatory ecological mechanisms.

### 4.3. Correlation Analysis

The existing correlation between physicochemical and microalgae abundance shifts has been extensively used as a water quality indicator [46]. Classic pollution-tolerant microalgae classification [47] identified the genera *Euglena*, *Chlamydomonas*, *Chlorella*, and *Ankistrodesmus* in the top ten most tolerant genera. These genera have been recently successfully employed as predictors of environmental health in river ecosystems [48]. In our case, taxa presence (for microscopic analysis) or the abundance (for metagenomic analysis) of *Euglena*, *Chlamydomonas*, *Chlorella*, and *Ankistrodesmus* genera also significantly correlated to different physicochemical indicators of water quality. However, seasonal variation in correlation significance has been reported [49], stressing the relevance of future work considering this variable. Our data come from the end of the dry season, expecting the highest concentration of pollutants and the highest degree of eutrophication; nevertheless, suggesting evidence of seawater intrusion in the case of the *Mocú* lagoon could alter seasonal responses.

### 4.4. Functional Analysis

Shotgun sequencing and gene annotation-derived data reveal the metabolic potential of eukaryotic microalgae communities in the Yucatan aquifer for the first time. To our knowledge, this is the first report using the metagenomic approach to characterize eukaryotic microalgal communities. Previous work on the functional assessment of these clades of microalga comes from axenic cultures of specific genera like *Chlamydomonas* [50], *Ankistrodesmus*, *Chlorella*, and *Scenedesmus* [51], and other ten freshwater and seawater genera compiled at the ALGAEFUN platform [52]. Unfortunately, beyond ITS metabarcoding markers, there are no genomic data for other genera to ease metagenomic analysis. This may explain the contrasting number of genera assigned in our samples compared to the limited number of annotated functions. This situation could be the case of dominant genera obtained from the bottom of the *Silvituc* lagoon, as *Ptilothamnion*, *Haramonas*, and *Micromonas* genera do not have genomic data reported, being limited only to ITS sequences, which correspond to the translation COG category. More categories were identified for those samples where genera with a genome sequence are available.

## 5. Conclusions

This work explores the eukaryotic microalgae community structure in three lagoons from the Yucatan aquifer. A high diversity of taxa were identified by combining microscopic and metagenomic approaches, 50% of which had not previously been identified at the genus level. In addition to the intensity of human activities associated with each particular lagoon, we could not correlate this environmental intensity use with the physicochemical parameters prevailing at each lagoon. In future studies, it will be necessary to include other parameters like biochemical oxygen demand and total suspended solids. Even some taxa of microalgae communities responded to varying parameters such as alkalinity, water hardness, cyanuric acid, carbonates, and lead; most taxa remained unaltered, suggesting that the resilient capability of the community remains. The most substantial difference in microalgae communities was found at the sample level, particularly for the community isolated from the bottom of the *Silvituc* lagoon, the most apparently perturbed site. We expected to find more taxonomic diversity in more conserved lagoons, but we could not find this correlation. Functional gene annotation reveals a similar pattern. Nonetheless, more evident differences were found in the diversity of identified COGs between lagoon origin, resulting in *Mocú*, the lagoon with more COG diversity. Further work on the genomics of rare microalgae clades would boost the identification of functional capabilities of biotechnological interest, such as water quality biomarkers or other applications.

## Figures and Tables

**Figure 1 microorganisms-12-02368-f001:**
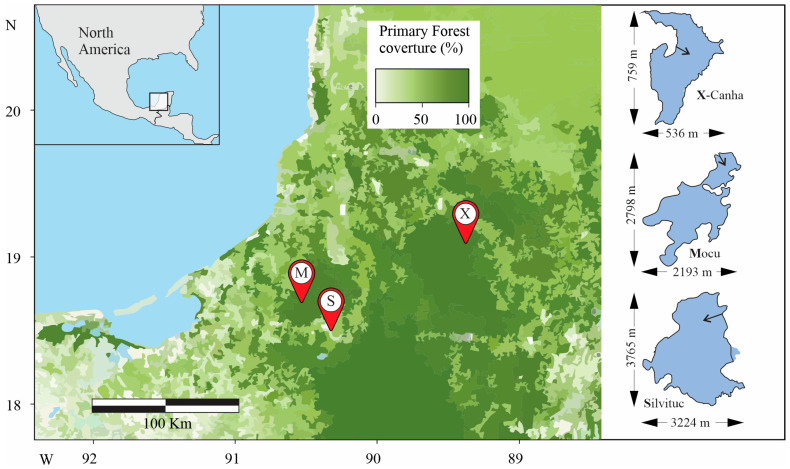
Sampling sites. The location of visited lagoons is shown on the left map, including the Forest Integrity [24] as a reference for environmental perturbation. In the left panel, the extension and shape of the lagoons. The arrow’s base in the lagoons corresponds to the dock location, and the tip corresponds to the inner sampling point.

**Figure 2 microorganisms-12-02368-f002:**
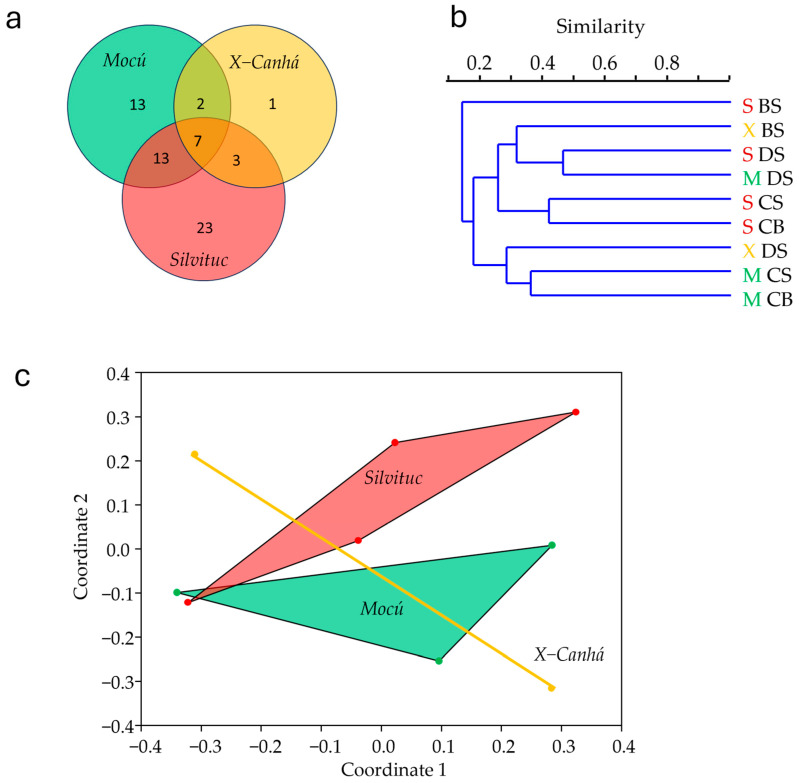
Microscopic analysis of microalgae community. (**a**) The Venn diagram shows the genus found at each lagoon; (**b**) Bray–Curtis clustering analysis. Green letter M corresponds to *Mocú* lagoon (pristine); yellow letter X to *X-Canhá* (semi-perturbed); and red letter S corresponds to *Silvituc* (perturbed). BS, bank surface; DS, dock surface; CS, center surface; and CB, center bottom; (**c**) non-metric dimensional scaling analysis showing convex hulls.

**Figure 3 microorganisms-12-02368-f003:**
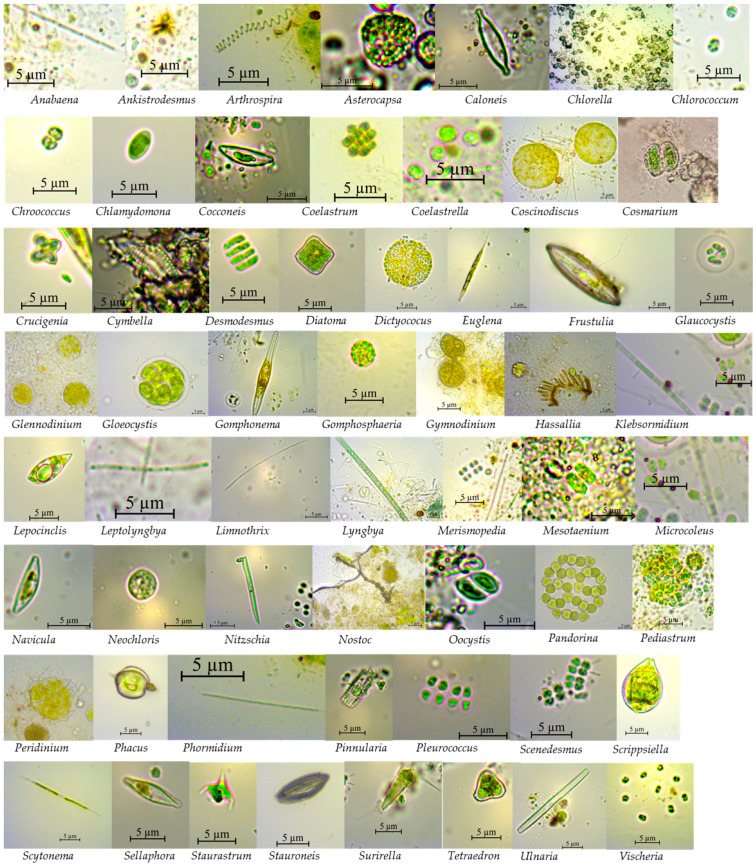
Eukaryotic microalgae specimens. Microscopic aspect of identified genera.

**Figure 4 microorganisms-12-02368-f004:**
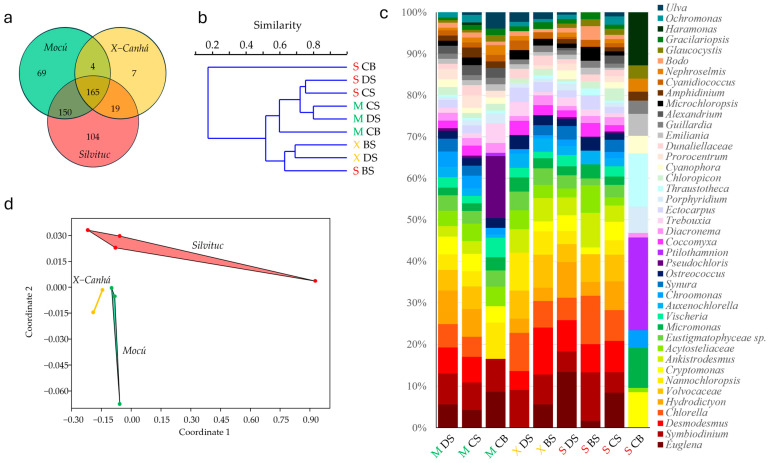
Metagenomic analysis of microalgae community. (**a**) The Venn diagram shows the genus found at each lagoon; (**b**) Bray–Curtis clustering analysis; (**c**) relative abundance of genera > 1% by sample; (**d**) non-metric dimensional scaling analysis showing convex hulls. Green letter M corresponds to *Mocú* lagoon (pristine); yellow letter X to *X-Canhá* (semi-perturbed); and red letter S corresponds to *Silvituc* (perturbed). BS, bank surface; DS, dock surface; CS, center surface; and CB, center bottom.

**Figure 5 microorganisms-12-02368-f005:**
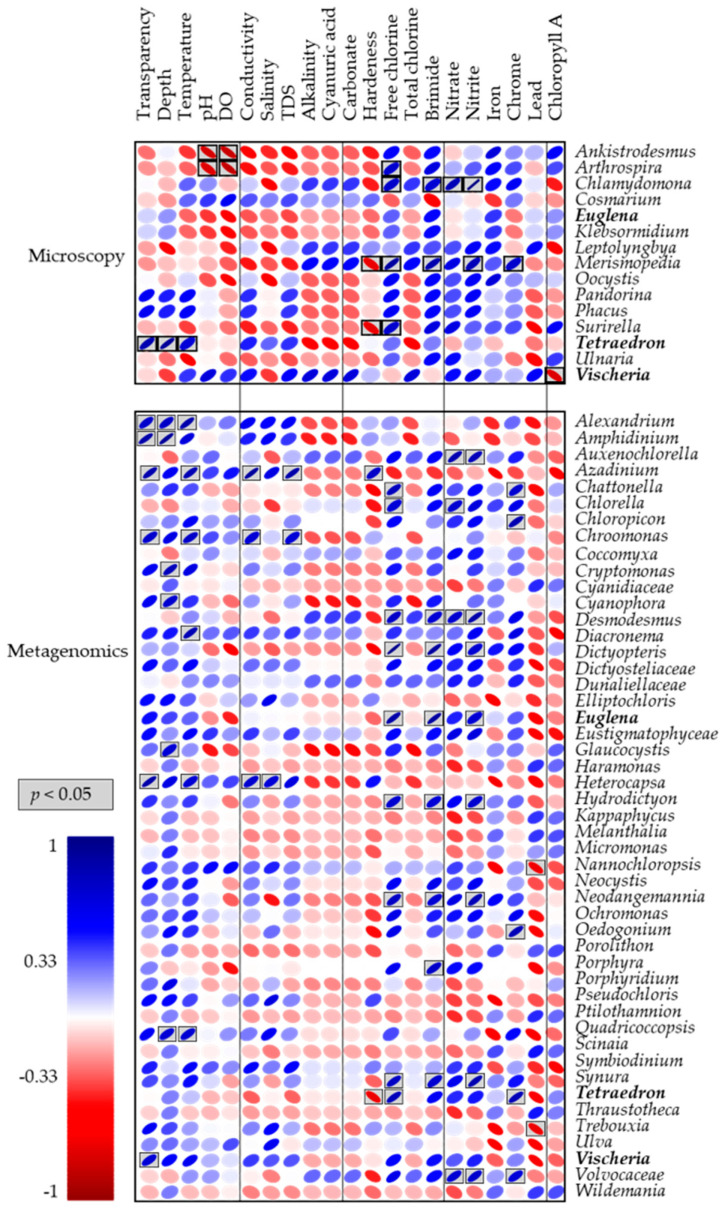
Pearson correlation analysis. Positive or negative correlations between environmental conditions and taxa presence (microscopic) or abundance (metagenomics) (*p* < 0.05). Genera in bold font corresponds to common genera identified by microscopic and metagenomic approaches.

**Figure 6 microorganisms-12-02368-f006:**
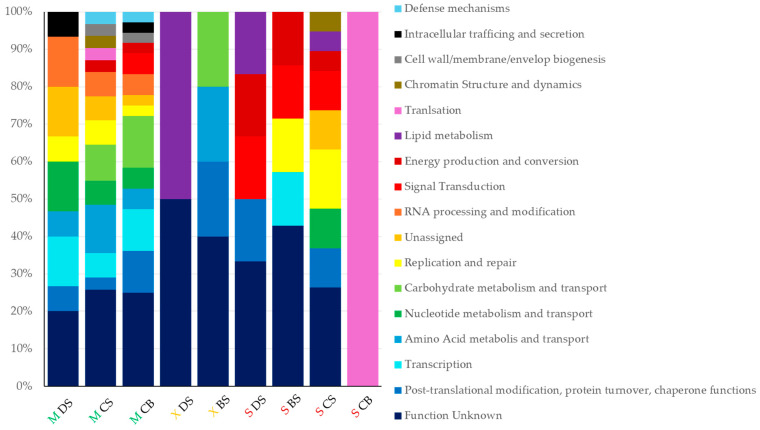
Functional gene annotation. Relative abundance by sample source of eukaryotic microalgae-filtered clusters of ortholog genes (COGs). Green letter M corresponds to *Mocú* lagoon (pristine); yellow letter X to *X-Canhá* (semi-perturbed); and red letter S correspond to *Silvituc* (perturbed). BS, bank surface; DS, dock surface; CS, center surface; and CB, center bottom.

**Table 1 microorganisms-12-02368-t001:** Water quality in sampling sites. DO: dissolved oxygen; TDS: total dissolved solids. Values correspond to the average value of three repetitions. Superscript letters at the right of numbers represent the differences in significance levels (*p* < 0.05).

Lagoon	Parameters/Sampling Site	Transparency (m)	Depth (m)	Temperature (°C)	pH	DO (mg L^−1^)	Conductivity (mS)	Salinity (ppm)	TDS (mg L^−1^)	Chlorophyll-a (mg L^−1^)
*Mocú*	Dock-Surface	1.3 ^bc^	1.4 ^ab^	32.5 ^d^	8.5 ^d^	6.4 ^cd^	2.17 ^e^	1002 ^c^	1570 ^e^	4.65 ^ab^
(Pristine)	Center-Surface	1.4 ^c^	2.5 ^c^	31.8 ^cd^	7.9 ^bcd^	6.2 ^bcd^	1.06 ^cde^	643 ^c^	663 ^cde^	6.42 ^abcd^
	Center-Bottom	1.4 ^c^	2.5 ^c^	31.8 ^cd^	7.9 ^bcd^	6.2 ^bcd^	1.13 ^de^	624 ^bc^	695 ^de^	5.67 ^abc^
*X-Canhá* (Semi-	Dock-Surface	0.5 ^ab^	0.5 ^ab^	29.5 ^a^	8.3 ^d^	6.6 ^d^	0.30 ^bcde^	144 ^bc^	186 ^bcde^	11.09 ^e^
perturbed)	Bank-Surface	0.4 ^a^	0.4 ^a^	30.8 ^b^	8.2 ^cd^	6.4 ^cd^	0.25 ^abcd^	116 ^ab^	170 ^abcd^	3.00 ^a^
*Silvituc*	Dock-Surface	1.2 ^bc^	1.6 ^abc^	30.3 ^ab^	7.5 ^ab^	3.1 ^a^	0.24 ^abc^	120 ^abc^	76 ^a^	7.22 ^abcde^
(Perturbed)	Bank-Surface	0.4 ^a^	0.4 ^a^	29.8 ^ab^	7.3 ^a^	3.8 ^ab^	0.22 ^a^	116 ^ab^	158 ^ab^	9.04 ^bcde^
	Center-Surface	0.7 ^abc^	2.0 ^bc^	31.5 ^c^	7.8 ^abc^	5.4 ^abc^	0.24 ^ab^	114 ^a^	185 ^abcde^	9.57 ^de^
	Center-Bottom	0.7 ^abc^	2.0 ^bc^	30.5 ^abc^	7.8 ^abc^	5.4 ^abc^	0.26 ^abcde^	116 ^ab^	168 ^abc^	9.51 ^cde^

**Table 2 microorganisms-12-02368-t002:** Water quality in sampling sites. Discrete values obtained from color strip tests. See Methods. ND. Not detectable.

Lagoon	Parameters (Range in mg L^−1^)/Sampling Site	Alkalinity	Cyanuric Acid	Carbonates	Hardness	Free Chlorine	Bromide	Nitrate	Nitrite	Iron	Chrome	Lead
*Mocú*	Dock-Surface	ND ^a^	ND ^a^	ND ^a^	425 ^b^	0.5 ^ab^	1.0 ^ab^	10 ^b^	1.0 ^b^	3.0 ^b^	ND ^a^	5.0 ^b^
(Pristine)	Center-Surface	ND ^a^	ND ^a^	ND ^a^	425 ^b^	ND ^a^	ND ^a^	ND ^a^	ND ^a^	ND ^a^	ND ^a^	ND ^a^
	Center-Bottom	ND ^a^	ND ^a^	ND ^a^	425 ^b^	ND ^a^	ND ^a^	ND ^a^	ND ^a^	ND ^a^	ND ^a^	ND ^a^
*X-Canhá* (Semi-	Dock-Surface	ND ^a^	ND ^a^	ND ^a^	250 ^ab^	ND ^a^	ND ^a^	10 ^b^	ND ^a^	1.5 ^ab^	ND ^a^	ND ^a^
perturbed)	Bank-Surface	180 ^b^	100 ^b^	180 ^b^	250 ^ab^	3.0 ^ab^	5.0 ^ab^	10 ^b^	1.0 ^b^	1.5 ^ab^	1.0 ^ab^	2.5 ^ab^
*Silvituc*	Dock-Surface	ND ^a^	ND ^a^	ND ^a^	250 ^ab^	10.0 ^b^	20.0 ^b^	10 ^b^	1.0 ^b^	3.0 ^b^	ND ^a^	ND ^a^
(Perturbed)	Bank-Surface	ND ^a^	ND ^a^	ND ^a^	425 ^b^	ND ^a^	ND ^a^	ND ^a^	ND ^a^	3.0 ^b^	ND ^a^	5.0 ^b^
	Center-Surface	ND ^a^	ND ^a^	ND ^a^	100 ^a^	10.0 ^b^	10.0 ^b^	10 ^b^	1.0 ^b^	3.0 ^b^	2.0 ^b^	ND ^a^
	Center-Bottom	ND ^a^	ND ^a^	ND ^a^	250 ^ab^	ND ^a^	ND ^a^	ND ^a^	ND ^a^	3.0 ^b^	ND ^a^	5.0 ^b^

**Table 3 microorganisms-12-02368-t003:** Photic zone and Attenuation coefficient.

Lagoon	Parameters/Sampling Site	Photic Zone Depth (m)	Attenuation Coefficient
*Mocú*	Dock-Surface	6.9188	0.6656
(Pristine)	Center-Surface	7.3462	0.6269
	Center-Bottom	7.3462	0.6269
*X-Canhá* (Semi-	Dock-Surface	6.5785	0.7
perturbed)	Bank-Surface	5.0044	0.9202
*Silvituc*	Dock-Surface	5.0044	0.9202
(Perturbed)	Bank-Surface	3.3998	1.3545
	Center-Surface	3.8999	1.1808
	Center-Bottom	3.3998	1.3545

## Data Availability

Shallow sequence data are available through the BioProject PRJNA1162095; https://www.ncbi.nlm.nih.gov/bioproject/1162095 (accessed on 4 September 2024).

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
