# Peer review of "Eukaryotic Microalgae Communities from Tropical Karstic Freshwater Lagoons in an Anthropic Disturbance Gradient Microscopic and Metagenomic Analysis"

_microorganisms, 2024, doi:10.3390/microorganisms12112368_

Round 1
Reviewer 1 Report
Comments and Suggestions for Authors
Paper is interesting and opens possibilities for deeper investigation. The main problem of the article is the taxonomical identification to the genus level. Identification to the species and variety levels is a must in investigations dealing with water quality. Genera like Chlorococcum, Chlamydomonas, Desmodesmus and some others have 100 or more species, and not all of them are tolerant to pollution. Please, try to identify species (at least) to get better conclusions.-
Comments on the Quality of English LanguageEnglish is alright, just with a few digitizing mistakes.-
Author Response
Comment 1. Paper is interesting and opens possibilities for deeper investigation. The main problem of the article is the taxonomical identification to the genus level. Identification to the species and variety levels is a must in investigations dealing with water quality. Genera like Chlorococcum, Chlamydomonas, Desmodesmus and some others have 100 or more species, and not all of them are tolerant to pollution. Please, try to identify species (at least) to get better conclusions.-
Response: We appreciate the comments and agree that identification at the species or variety level provides further details on some genera. However, this work is not focused on taxonomical identification at the deepest level but on a taxonomic depth that allows us to classify most organisms found to analyze the microalgae community as a whole. Further analysis of culturable microalgae could lead to more precise taxonomic identification of particular taxa.
Reviewer 2 Report
Comments and Suggestions for Authors
The authors have presented an interesting metagenomics survey of lagoons in Mexico while looking for indications of human influence on natural ecosystems.
Major Comments:
- The results and discussion should be revised to make their main points better justified. In the introduction, they should clearly state their hypothesis or research question about what they thought the influence from human activities would be before they conducted the study. Then in the results and discussion- What is the evidence of anthropogenic influence in natural systems or not? Do their results confirm or contradict their hypothesis? In the discussion, what are the implications of their findings for these systems? Does it imply that natural ecosystems are not functioning as they used it? How have natural microbial communities been disturbed?
- They should also describe more about how the new types of microalgae were identified and why these would be important. They describe the results fine but don’t discuss much about what the implications are for their system or how these communities compare to other related environments that have been studied in other regions?
Specific comments:
- Details in methods should be expanded. Especially 2.4 and 2.5 for metagenomic and bioinformatic analyses. Much more information on how samples were extracted and analyzed is needed. The authors report the exciting finding of over 50 new identifications and much more information on how that was determined is needed.
- What is the level of replication at each site?
- In 2.1- they say the type of bottle used in the sampling but how did they actually collect the water? With a hose or bucket etc?
- Likewise in 2.1 and Table 1 more information should be included on how they measured the water quality parameters- which specific probes and test strips etc were used? Were they used in the field? Were technical replicate measurements conducted?
Comments on the Quality of English LanguageA few minor typos to be corrected. For example, in Figure 3 caption, 'vent' diagram is incorrect.
Author Response
Comment 1 The results and discussion should be revised to make their main points better justified. In the introduction, they should clearly state their hypothesis or research question about what they thought the influence from human activities would be before they conducted the study. Then in the results and discussion- What is the evidence of anthropogenic influence in natural systems or not? Do their results confirm or contradict their hypothesis? In the discussion, what are the implications of their findings for these systems? Does it imply that natural ecosystems are not functioning as they used it? How have natural microbial communities been disturbed?
Response: We have added specific information regarding these questions in the results and discussion sections, lines 160-163; 303-306, and 321-324.
Comment 2: They should also describe more about how the new types of microalgae were identified and why these would be important. They describe the results fine but don’t discuss much about what the implications are for their system or how these communities compare to other related environments that have been studied in other regions?
Response: This article analyzes the eukaryotic microalgae community. Some relevant taxa are discussed, but we consider that an extensive discussion of all new taxa found would substantially increase the length of this paper and disrupt the discourse focused at the community level. Regarding the comparison to other regions, we included several works (Line 328, cites 10, 32, 33; Line 333, cites 9, 35; Line 357, cite 41).
Comment 3: Details in methods should be expanded. Especially 2.4 and 2.5 for metagenomic and bioinformatic analyses. Much more information on how samples were extracted and analyzed is needed. The authors report the exciting finding of over 50 new identifications and much more information on how that was determined is needed.
Response: We provide standard information for DNA extraction and the bioinformatic pipeline and the supporting cites for both processes. The information already provided should lead to an average bioinformatician repeating the bioinformatic process.
Comment 4. What is the level of replication at each site?
Response: "Each composite sample corresponded to a mix of seven water samples" has been added to line 86.
Comment 5. In 2.1- they say the type of bottle used in the sampling but how did they actually collect the water? With a hose or bucket etc?
Response: An explanation was included in line 91. Direct transfer.
Comment 6. Likewise in 2.1 and Table 1 more information should be included on how they measured the water quality parameters- which specific probes and test strips etc were used? Were they used in the field? Were technical replicate measurements conducted?
Response: The requested information is presented in lines 97-102. Measurement repetitions have also been added at line 97.
"Venn" has been corrected in captions from figures 3 and 4.
Reviewer 3 Report
Comments and Suggestions for Authors
Very well written can be accepted in the present form
Author Response
No comments
Reviewer 4 Report
Comments and Suggestions for Authors
I have reviewed the manuscript entitled “Eukaryotic Microalgae Communities from Tropical Karstic Freshwater Lagoons in an Anthropic Disturbance Gradient. Microscopic and Metagenomic Analysis”. This manuscript is well written and effectively delivers its findings. The rationale for performing the study is laid out comprehensively. However, some parts of the manuscript required further elaboration. In addition, the manuscript would benefit from a thorough grammatical revision, ensuring a unified and fluid language that is understandable to the reader.
Below I present some suggestions that could help improve the presentation of the manuscript.
Line 43. Cyanobacteria, the dominant species or at least genera should be included.
Line 78. In this section of the manuscript, it would be desirable to present a hypothesis; indeed, it could be related to the degree of perturbation of each environment, how would Eukaryotic microalgae communities be expected to respond?
Line 85. Indicate the total depth of each sampling site. It would also be desirable to include information on the region's climatology, how it changes between the different climatic seasons (it can be between dry and rainy), and how the hydrography of each lagoon changes, for example. This can give the reader a more detailed overview of the sampling sites. The above considerations should be considered when interpreting and discussing the results.
Line 90. Indicate whether the Van Dorn bottle was rinsed with an acid solution or distilled water between each sample to avoid contamination.
Line 92. “chlorophyll-A” must be chlorophyll-a
Line 94. using the Secchi disk the authors can estimate the photic layer as well as the attenuation coefficient; they can use these parameters to correlate with chlorophyll-a
Line 97. “Total Dissolved Solids (TDS)” > total dissolved solids.
Line 101. Were the sensors calibrated prior to each sampling? Please indicate. In the case of nutrients, what was the sensitivity of the analyses? Please indicate.
Figure 1. The right panel indicates the length of the water body, but what is the width?
Line 108. At what pressure (psi) were the samples filtered?
Line 107. “chlorophyll A” should be chlorophyll-a
Line 117. Idem.
Line 118. Idem.
Line 155. There is no Table 2.
Line 157. “statistical differences were found between lagoons for some parameters” What was the value of r, or p? What kind of statistical tests were applied to check if the differences were statistically significant. Did you use Mantel or Pearson’s correlation coefficient tests or which one?
Line 160. “chlorophyll-A” should be chlorophyll-a
Line 162. “Superinex letters” > superscript letters?
Table 1. Many of the chemical parameters are reported with a value of 0, was it really 0 or was it undetectable? This depends on the sensitivity of the analyses, so it is necessary to indicate the sensitivity of each analysis in the materials and methods section. On the other hand, it is not really clear what the superscript letters (e.g. a, b, etc) indicate, for example the letter a denotes an environment with greater disturbance? Please clarify this from the Materials and Methods section.
Figure 2. Since in all cases the scale bar is presented at 5 µm, I recommend that only one bar be placed, so that the size of each figure could be increased. In fact, an index (a), b), c), etc.) could be included for each genus and thus save space.
Line 238-239. Negatively? Or Inversely?
Line 288-290. This is true, the “cenotes” have a connection with the sea, mixing their waters and changing the hydrographic properties.
Line 300. What about the “Nortes” period that impacts the region. Will there be any response in this type of environment?
Finally, I think that the Discussion section could be considerably improved if the authors consider the climatic variability of the region, which is extremely dynamic and therefore promotes great variability in organisms at all levels. I understand that there are few works on this subject, but the authors could make an effort and propose some scenarios; I think that it could be a good contribution.
Comments on the Quality of English LanguageModerate editing of English language required.
Author Response
Thank you very much for your review and comments!
Comment 1) Line 43. Cyanobacteria, the dominant species or at least genera should be included.
Response: Done. Dominant genera from three works from tropical karstic environments have been added.
Comment 2): Line 78. In this section of the manuscript, it would be desirable to present a hypothesis; indeed, it could be related to the degree of perturbation of each environment, how would Eukaryotic microalgae communities be expected to respond?
Response: Done.
Comment 3) Line 85. Indicate the total depth of each sampling site. It would also be desirable to include information on the region's climatology, how it changes between the different climatic seasons (it can be between dry and rainy), and how the hydrography of each lagoon changes, for example. This can give the reader a more detailed overview of the sampling sites. The above considerations should be considered when interpreting and discussing the results.
Response: Done. A paragraph has been added in section 2.1
Comment 4) Line 90. Indicate whether the Van Dorn bottle was rinsed with an acid solution or distilled water between each sample to avoid contamination.
Response. Done. An explanation of the sanitization process has been included.
Comment 5) Line 92. “chlorophyll-A” must be chlorophyll-a
Response. Corrected.
Comment 6) Line 94. using the Secchi disk the authors can estimate the photic layer as well as the attenuation coefficient; they can use these parameters to correlate with chlorophyll-a
Response: Done. Both indices were calculated and incorporated into Table 1.
Comment 7) Line 97. “Total Dissolved Solids (TDS)” > total dissolved solids.
Line 101. Were the sensors calibrated prior to each sampling? Please indicate. In the case of nutrients, what was the sensitivity of the analyses? Please indicate.
Comment 8) Figure 1. The right panel indicates the length of the water body, but what is the width?
Response. Fixed. Now, Figure 1 shows the width value of each lagoon
Comment 8) Line 108. At what pressure (psi) were the samples filtered?
Response: Clarified.
Comment 9) Line 107. “chlorophyll A” should be chlorophyll-a
Response: Done. Corrected.
Line 117. Idem.
Response: Done. Corrected.
Line 118. Idem.
Response: Done. Corrected.
Comment 9) Line 155. There is no Table 2.
Response. Corrected (It was an error; Table 2 is Table 1 continued)
Comment 10) Line 157. “statistical differences were found between lagoons for some parameters” What was the value of r, or p? What kind of statistical tests were applied to check if the differences were statistically significant. Did you use Mantel or Pearson’s correlation coefficient tests or which one?
Response: Corrected. The M&M section (Lines 110-121) and results section (lines 171-180) were re-written
Comment 11) Line 160. “chlorophyll-A” should be chlorophyll-a
Response: Corrected
Comment 12) Line 162. “Superinex letters” > superscript letters?
Response: Corrected
Comment 13) Table 1. Many of the chemical parameters are reported with a value of 0, was it really 0 or was it undetectable? This depends on the sensitivity of the analyses, so it is necessary to indicate the sensitivity of each analysis in the materials and methods section. On the other hand, it is not really clear what the superscript letters (e.g. a, b, etc) indicate, for example the letter a denotes an environment with greater disturbance? Please clarify this from the Materials and Methods section.
Response: Corrected. The M&M section (Lines 110-121) and results section (lines 171-180) were re-written
Comment 14) Figure 2. Since in all cases the scale bar is presented at 5 µm, I recommend that only one bar be placed, so that the size of each figure could be increased. In fact, an index (a), b), c), etc.) could be included for each genus and thus save space.
Response: We appreciate your comments; however, altering the photo size will lead to pixelization because our microscope/camera is not very good quality. Moreover, we consider the figure more manageable to read if the name is below each photograph instead of using letters.
Comment 15) Line 238-239. Negatively? Or Inversely?
Response: Corrected.
Comment 16) Line 288-290. This is true, the “cenotes” have a connection with the sea, mixing their waters and changing the hydrographic properties.
Response: Yes, we agree. However, as this fact has not been demonstrated for this particular lagoon, and our data does not prove that, we opt for using a cautious discourse.
Comment 17) Line 300. What about the “Nortes” period that impacts the region. Will there be any response in this type of environment?
Response: Some lines have been added in the discussion section (Lines 321-325)
Comment 18) Finally, I think that the Discussion section could be considerably improved if the authors consider the climatic variability of the region, which is extremely dynamic and therefore promotes great variability in organisms at all levels. I understand that there are few works on this subject, but the authors could make an effort and propose some scenarios; I think that it could be a good contribution.
Response: A paragraph has been added at the end of the discussion section 4.2 (Lines 376-381)

Round 2
Reviewer 4 Report
Comments and Suggestions for Authors
After carefully reviewing the new submitted version and going through the responses point by point, I identified that the authors worked hard to satisfactorily respond to my suggestions, which I greatly appreciate.
I now find a better organized manuscript, with a high quality presentation and understandable for the reader.
Therefore, I suggest accepting this version and continuing with the editing. In my opinion, it is an excellent contribution.
Well done!